# Synthetic Nucleic Acid Antigens in Localized Scleroderma

**DOI:** 10.3390/ijms242417507

**Published:** 2023-12-15

**Authors:** Sangita Khatri, Adrian H. Bustos, Christian Damsgaard Jørgensen, Kathryn S. Torok, Lise-Mette Rahbek Gjerdrum, Kira Astakhova

**Affiliations:** 1Department of Chemistry, Technical University of Denmark, 2800 Kongens Lyngby, Denmark; khatrisangita2049@gmail.com (S.K.); adrianb@kemi.dtu.dk (A.H.B.); 2Department of Mathematical Sciences, Aalborg University, 9220 Aalborg, Denmark; cdjo@math.aau.dk; 3Department of Mathematics and Computer Science, University of Southern Denmark, 5230 Odense, Denmark; 4Department of Pediatrics, UPMC Children’s Hospital of Pittsburgh, Pittsburgh, PA 15224, USA; 5Department of Pathology, Zealand University Hospital, 4000 Roskilde, Denmark; 6Department of Clinical Medicine, University of Copenhagen, 2100 Copenhagen, Denmark

**Keywords:** localized scleroderma, synthetic nucleic acid antigens, dsD4, autoimmunity

## Abstract

We investigated the impact of synthetic nucleic acid antigens on the autoantibody profiles in patients with localized scleroderma, an autoimmune skin disease. Anti-DNA antibodies, including double-stranded DNA (dsDNA) and single-stranded DNA (ssDNA), are common among autoimmune diseases, such as systemic lupus erythematosus and localized scleroderma. Based on recent studies, we hypothesized that the sequence of nucleic acid antigens has an impact on the autoimmune reactions in localized scleroderma. To test our hypothesis, we synthesized a panel of DNA and RNA antigens and used them for autoantibody profiling of 70 children with localized scleroderma compared with the healthy controls and patients with pediatric systemic lupus erythematosus (as a disease control). Among the tested antigens, dsD4, which contains the sequence of the human oncogene *BRAF*, showed a particularly strong presence in localized scleroderma but not systemic lupus erythematosus. Disease activity in patients was significantly associated with dsD4 autoantibody levels. We confirmed this result in vivo by using a bleomycin-induced mouse model of localized scleroderma. When administered intraperitoneally, dsD4 promoted an active polyclonal response in the mouse model. Our study highlights sequence specificity for nucleic acid antigens in localized scleroderma that could potentially lead to developing novel early-stage diagnostic tools.

## 1. Introduction

Antinuclear antibodies (ANAs) have been known for over four decades. In recent years, there has been increased attention toward ANAs regarding diagnosing and monitoring the progression of autoimmune diseases; indeed, it is a core criterion for the diagnosis of systemic lupus erythematosus (SLE) [1]. ANAs are also common, but not a diagnostic criterion, in autoimmune skin diseases such as localized scleroderma (LS), also termed morphea, and systemic scleroderma, also called systemic sclerosis (SSc) [1].

LS is a complex disease with diverse clinical manifestations; it is mainly characterized by collagen deposition limited to the skin and underlying connective tissue, but there may also be systemic involvement [2]. LS and SSc both have the same skin histopathological changes, including an inflammatory infiltrate in the superficial and reticular dermis, typically surrounding vessels and skin appendages, along with an expansion of collagen with thickened collagen fibrils, but these conditions have different extracutaneous and internal organ manifestations. However, LS is 10 times more prevalent than SSc and lacks a serological diagnosis and clear knowledge of its pathogenesis and management [3]. Some of the larger cohort studies have shown elevated ANA levels in patients with LS: 23–68% of patients are ANA-positive, including a nearly 50% positivity for positive anti-single-stranded DNA (ssDNA) antibodies [4,5]. Furthermore, the positivity for anti-histone (AHA) and anti-ssDNA antibodies has been associated with more severe LS with deep tissue involvement and joint contracture [4,5].

Cytokines, including interleukin (IL)-2, IL-4, IL-6, IL-8, IL-13, and tumor necrosis factor alpha (TNF-α), and autoantibodies are related to LS pathogenesis [6]. Inflammation is caused by an imbalance in the Th1, Th2, and Th17 cell subsets in the early stages of the disease; Th2 cells contribute to fibrosis during the second stage [7]. A better understanding of the active inflammatory phase in LS could improve the understanding of the basic biology and lead to direct and efficient therapies. The current evidence points toward the association between specific human leukocyte antigen (HLA) class I and II alleles in patients with LS that are different from those found in patients with SSc. This feature makes LS distinct [8].

The role of HLA molecules has been well described. LS-specific HLA profiles can lead to B cells producing certain cytokines and autoantibodies, which contribute to disease progression [9,10]. RNA sequencing of inflammatory lesioned LS skin has revealed an increased expression of chemokine (C-X-C motif) ligand genes, including *CXCL9*, *CXCL10*, and *CXCL11*, and their receptor *CXCR3*, with cell-specific transcripts of infiltrating macrophages and T cell subsets [11]. The underlying pathogenesis is triggered by altered collagen production by a combination of vascular damage, infiltration of T cells that release IL-4, and transforming growth factor beta (TGF-β)-activated fibroblasts.

Besides phenotypic changes, LS is associated with major shifts in gene expression (Appendix A) [11,12]. The affected genes include CD receptor genes, cell adhesion genes, chemokines, the interferon-stimulating gene *ISG5*, genes involved in regulatory pathways (*KRAS*, *LYN*, and *BLK*), and the B-cell-regulating gene *MZB1*. Fibrinogen and collagen genes are also affected. In addition, *BAX* and *BCL2*, which regulate T cells and apoptosis, respectively, are also affected in LS.

RNA sequencing of a bleomycin-induced mouse model of LS and of human samples has indicated that the TNF-related apoptosis-inducing ligand (TRAIL) pathway plays a major role in the disease [13]. The authors found upregulated *DR4*, *DR5*, *ACTA2*, *COLLA2*, *TGFB1*, and *PDGFRB* messenger RNA (mRNA) in skin biopsies from patients with LS. Thess changes increase caspase-8 and caspase-3/7 activity in skin. The authors detected mRNA of these genes in fibrotic skin from LS mice and patients with LS. Human TRAIL is active in mice, which allowed the authors to confirm the upregulation of genes in the animal model as well. Based on the results, DR5 is a potential therapeutic target for LS. Of relevance to in vivo studies, there is only partial conservation of the SLE genetic signature in mice. Approximately 20% of the genes involved in SLE in humans are conserved in mice. There have been few genomic studies regarding LS mouse models. They have revealed about a 30% conservation with the human genetic signature of LS (Appendix A).

The use of different oligonucleotide antigens for diagnostics and gene therapy of autoimmune diseases has been poorly explored. Nevertheless, synthetic oligonucleotides with high specificity and affinity and controlled purity and chemical content have proven to be a promising tool for diagnostics and studies of autoimmune diseases. In our previous study, we investigated sequence-controlled rationally designed DNA, RNA, and locked nucleic acid (LNA) antigens in SLE. This approach might allow us to investigate the details on the antigen recognition by autoimmune antibodies in LS [14,15]. Supported by the recent findings we obtained by using sequence-controlled antigens and human serum samples [14,15], we hypothesized that the nucleotide sequence of nucleic acid antigens has an impact on the autoimmune reactions in LS. We tested our hypothesis in vitro and in vivo using human samples and a mouse model of LS and compared it to SLE as a disease control and to healthy controls. To the best of our knowledge, this is the first study that has investigated the sequence specificity of nucleic acid antigens in LS in human samples and a mouse model of the disease.

## 2. Results and Discussion

First, we evaluated the autoantibody profiles of sequence-defined antigens in human samples. We used the sequences of antigens from our previous work and included ssDNA, ssRNA, double-stranded DNA (dsDNA), and dsRNA (Figure 1a). The D4 and D5 DNA antigens had been applied successfully in an SLE study [16]. There was elevated LNA-modified dsLD3 in the patients with LS. dsHUV is derived from human cytomegalovirus and is a reactive antigen for autoantibodies in patients with SLE and neuropsychiatric SLE [16]. We included RNA variants of the antigens D4, D5, and RNA of the Epstein–Barr virus in the panel of antigens based on previous reports on RNA antigenicity. RNA variants are more abundant in cells and tissues, although they are less enzymatically stable. We included calf thymus dsDNA (CTD) as a control for specificity. We compared the enzyme-linked immunosorbent assay (ELISA) results for serum from patients with LS, SLE, and healthy controls with Tukey’s honest significant difference test to control the family-wise error rate at the 0.05 level.

The ELISA results and their statistical assessment are shown in Figure 1 and the Appendix A. The immunoglobulin G (IgG) and immunoglobulin M (IgM) tests revealed similar autoantibody recognition patterns in the LS cohort for various DNA antigens and significant differences between the LS, SLE, and healthy control cohorts (Figure 1b,c). The IgM levels were higher than the IgG levels and significantly different between the study groups. For dsD4, the IgM levels were particularly high (Figure 1c), pointing to nonspecific binding. We also tested for IgE antibodies and observed elevated levels in the SLE and LS cohorts (Figure 1d). For SLE, this is in accordance with previous reports on IgE ANAs observed in about 50% of patients with SLE [17]. Overall, the IgG antibodies to DNA antigens showed specificity for LS. In turn, the antibodies to RNA were elevated in the healthy controls, in agreement with previous reports (Appendix A) [14,15].

We fitted a covariance generalized linear model (cGLM) [18] with the logit link function and the binomial variance function to the IgG ELISA data for the patients with LS. We used the compound symmetry covariance structure to account for a within-patient correlation induced by repeated measurements (Appendix A). Interestingly, the model suggests that the disease score was highly significantly affected by dsD4 and ssD4 (Appendix A).

Next, we conducted an in vivo study using a bleomycin-induced LS mouse model and a tamoxifen-sensitized NZB/NZW SLE mouse model [19,20,21,22]. Based on the serology results of the human samples, we selected the two most potent antigens for the in vivo study: dsD4 and dsD5. There is a confirmed association between dsD4 and LS disease activity, and dsD5 is specific for SLE [14,15]. dsD4 is a sequence fragment of the human oncogene *BRAF*; it is a mixmer oligonucleotide with a 43% GC content. dsD5 is based on the TC dinucleotide that was originally used to develop highly reactive monoclonal antibodies to DNA.

We expected that the DNA antigens and their corresponding autoantibodies would already be present in the LS and SLE mice as they developed the disease. The idea behind our experiment was to evaluate whether a high level of exogeneous antigen and the autoantibodies produced toward this exogeneous antigen would be associated with the disease course and the autoantibody profiles. T cell presentation is required to develop autoantibodies. Dendritic cells can present exosomal DNA to T cells [23]. Moreover, Luo et al. [24] reported that cytosolic dsDNA induces major histocompatibility complex (MHC) class I molecules due to an autocrine/paracrine effect of type I interferon (IFN). This suggests that the exogenous dsD4 and dsD5 can induce autoantibodies and impact the course of LS.

Figure 2 shows the in vivo study scheme and the obtained readouts. To induce LS, we treated C57BL/6J mice with a single intraperitoneal dose of bleomycin at 25 mg/kg body weight. This is a rather mild induction compared with reported daily bleomycin doses of 50–100 mg/kg body weight [25,26]. Our goal was to produce a localized disease model, with the main manifestation of LS in skin and little to no systemic involvement. Similarly to using SLE as a disease control in the experiment with human serum, we also used a tamoxifen-sensitized NZB/NZW F1 model of SLE as a disease control group for the in vivo investigation. We administered tamoxifen intraperitoneally three times at 50 mg/kg body weight per dose. We used healthy C57BL/6J mice as controls (Figure 2B). After induction, we treated the mice once a week with a high or low intraperitoneal dose of DNA antigens (14 and 1.4 µg/dose, respectively). A control group received phosphate-buffered saline (PBS). The animals received three total treatments over the course of 3 weeks. During the observation, we scored the mice for clinical manifestations twice a week, and we measured weight and collected blood (tongue) samples once a week. There were five mice per group. Using a post hoc power calculation with an expected mean difference in the ELISA autoantibody levels >0.05 absorbance units (expected value range 0.1–1.0), the power of the study was 97% [27].

All the mice showed good condition (Appendix A). After induction of the disease, the NZB/NZW mice showed a faster growth rate than the C57BL/6J mice (Appendix A). The skin condition, food intake, and activity of all the animals was good during the entire study.

We analyzed serum samples from the mice. We collected the serum samples eight times and analyzed them by using in-house ELISAs for dsD4, dsD5, dsHUV, ag1, and PLox. We included oxidized phospholipids (PLox), both IgG and IgM, to assess potential inflammation and oxidative stress as a result of the antigen administration [28,29]. We also accounted for the impact of the antigen dose and sequence. Based on one-way analysis of variance (ANOVA), there was at least one significant difference in the autoantibody levels during the course of the study. We terminated the study after 5 weeks (Figure 2).

We analyzed autoantibody levels at two times: at the end of the antigen administration and at the end of the observation (Figure 2). The complete data can be found in Appendix A. The results for the end of the observation and the high antigen dose are shown in Figure 3. In general, we observed a polyclonal autoantibody response to both antigens in all the animal groups, with some differences between IgG and IgM and between the animal groups. A disbalance in IgG degradation–stabilization mediated by a low-affinity polyclonal IgM response has been connected to the onset and progression of autoimmunity, and it stands as a potential therapeutic target [30].

For the LS mice, the administration of dsD4 resulted in similar levels of IgG autoantibodies to all the analyzed nucleic acid antigens (Figure 3a). However, the administration of dsD5 led to higher levels of anti-dsD5 compared with the other autoantibodies. The IgG levels of only half of the analyzed antigens responded to the higher antigen dose (Appendix A). At the end of the observation, on average, the anti-dsD4, anti-dsHUV, and anti-PLox levels in the LS mice sera had increased by 1.8%, 3.5%, and 10%, respectively, compared with the end of the treatment. The anti-dsD5 IgG levels did not change over the course of the study.

In the LS mice, the serum IgM autoantibody levels increased at the higher antigen dose (Appendix A). The administration of dsD4 or dsD5 led to elevated levels of polyclonal autoantibodies to nucleic acid antigens (Figure 3b). In contrast to the IgG levels, the IgM levels decreased at the end of the observation compared with the end of the treatment. This was the case for all the IgM autoantibodies against nucleic acid antigens, with an average decrease of 3–4.5%. Only anti-PLox IgM increased slightly (by 1.3%) in the LS mice.

After the antigen administration, the unimmunized mice did not show signs of scleroderma development in the skin. However, they developed autoantibodies. For IgG, this was particularly observed for the anti-RNA and anti-PLox levels, pointing toward an inflammatory reaction (Figure 3a). For all the other nucleic acid antigens, the IgG autoantibody levels decreased by 3.0–4.5% over the course of the study.

The IgM response in the unimmunized mice treated with dsD4 or dsD5 showed the same trend as in the immunized LS mice (Appendix A; Figure 3b). The antigen dose influenced the autoantibody levels, which decreased by 3.4–5% at the end of the observation for all the nucleic acid antigens but increased by 2.4% for anti-PLox. Both the LS and healthy mice showed the highest IgG and IgM levels to PLox, pointing toward an inflammatory response due to LS in the induced mice and antigen administration (Figure 3).

As a control, we analyzed the serum autoantibody profiles in tamoxifen-induced NZB/NZW mice. As expected, the autoantibody levels in the SLE mice were nearly 2-fold higher compared with the LS mice (Appendix A; Figure 3). The antigen had a strong effect on the IgG and IgM autoantibody levels. The anti-RNA levels were lower in the SLE mice compared with the LS and healthy mice. The levels of autoantibodies to the other nucleic acid antigens were similar after the administration of dsD4 or dsD5. Differently from the LS and healthy mice, the anti-PLox IgG levels in the SLE mice did not change much at the end of the observation.

The serology data support our previous in vitro results regarding the sequence specificity of the antibody response to nucleic acids [15,16,17]. The interaction between nucleic acids and B cell receptors and the subsequent activation of autoimmune responses has been reported [31]. Thus far, mechanistic details have been provided for a broad range of antigens, mostly focusing on toll-like receptors and pathogenic antigens. When the interaction with B cell receptors is antigen-specific, it can lead to a sequence-dependent immune response. This phenomenon is supported by our data. We also observed that the autoantibodies produced in the mice as a response to the administration of our antigens were polyclonal. To the best of our knowledge, this has not yet been described in the context of LS.

## 3. Materials and Methods

### 3.1. Patients and Controls

This retrospective serology study used samples from patients with LS, patients with SLE, and healthy controls. See Appendix A for complete data on the subjects.

Clinical measures and outcome data associated with the samples were extracted from the NRCOS registry. Demographic variables included sex, race, and age at the visit when the sample was collected. Healthy controls were age- and sex-matched at a 1:1 ratio. Additional clinical variables for patients with LS included validated measures of disease activity and severity, including the modified localized scleroderma skin index (mLoSSI), which quantifies cutaneous disease activity. The mLoSSI and the physician global assessment of activity (PGA-A) are the core variables defining disease activity in LS and have been found to be responsive to change [11]. The PGA-A includes consideration of the following cutaneous variables: new lesions within the previous month, erythema/violaceous color at the border of the lesion, and skin thickening/induration at the border of the lesion. Patients with an mLoSSI score > 3 and a PGA-A score >10 were considered to have active disease. Patients with mLoSSI and PGA-A scores of 0 were considered to have clinically inactive LS.

Patients with SLE were matched to the LS cohort at a 1:1 ratio. The cohort was 73% females, with a median age at sample collection of 14.6 years. Forty percent of the patients with SLE were in remission. Matched healthy controls were young adults with a median age of 19 years (73% females).

### 3.2. In-House ELISA of Human Samples

The antigens were synthesized in-house and characterized by HPLC and MALDI MS as described previously [14,15]. The general protocol for the in-house ELISA is given below.

Antigens: CTD from Sigma-Aldrich (cat no. 2618; Copenhagen, Denmark) was used as a specificity control. Single-stranded antigens were used without prior annealing. Each well of the ELISA plate included an ss/ds DNA, LNA, or RNA antigen, or CTD. For duplexes, single-stranded complementary strands were annealed at a 1:1 molar ratio in 10× PBS (Sigma-Aldrich, Copenhagen, Denmark) for 10 min at 92 °C, followed by cooling to room temperature for 30 min. The volume was adjusted with 1× PBS to obtain a final antigen concentration of 3.5 µg/mL.

Secondary antibodies (conjugated antisera): Species-specific (human and mice) IgG (Fc specific), IgM (µ-chain specific), IgA, and IgE (ε-chain specific) developed in goat or rabbit were used as secondary antibodies (Sigma-Aldrich, Copenhagen, Denmark). Horseradish peroxidase (HRP)-conjugated antibody solution was prepared by diluting commercially available anti-serum conjugate in previously prepared diluent (2 g bovine serum albumin (BSA, Sigma-Aldrich, Copenhagen, Denmark), 50 µL Tween-20 (Sigma-Aldrich, Copenhagen, Denmark), and 1 L 1× PBS) at a ratio of 1:20,000.

For the ELISA, Maxisorb 96-well plates (Nunc, Copenhagen, Denmark) were coated with antigens (one per well) at a concentration of 3.5 µg/mL in 1× PBS (100 µL/well) and incubated at room temperature overnight (12–18 h). The plates were blocked with 1× PTB buffer (20 g BSA, 50 µL Tween-20, and 1 L 1× PBS; 150 µL/well) at 37 °C. Serum was diluted at a ratio of 1:100 in diluent (2 g BSA, 50 µL Tween-20, and 1 L 1× PBS; 100 µL/well), and the plate was incubated for 1.5 h at 37 °C. After four washes with 1× PBS, 100 µL of HRP-conjugated secondary antibody was added to each well; the plate was incubated at 37 °C for 1.5 h. After four washes, freshly prepared TMB solution (Sigma-Aldrich, Copenhagen, Denmark) was added to each well (3 mg of TMB, 5 mL of DMSO diluted to 50 mL with 0.1 M acetate buffer, pH 5.4–5.7, and 3 µL H_2_O_2_; 100 µL/well), and the plate was incubated for 20–30 min at room temperature. The reaction was stopped by adding 1 M H_2_SO_4_ (50 µL/well). The absorbance at 450 nm was measured in each well. Statistical analyses were carried out in R (version 4.3.1, R Core Team, R Foundation for Statistical Computing, Vienna, Austria). A *p*-value less than 0.05 was considered statistically significant.

### 3.3. Non-Induced Mice

Eight-week-old mice were purchased from Charles River, Freiburg, Germany, and acclimatized for 5 days. The average weight was 18 g. C57BL/6J mice were treated with intraperitoneal injections of dsD4 or dsD5 (14 or 1.4 µg/dose). Antigens were freshly annealed in 1× PBS and tested to ensure they were LPS-free by using an LPS kit from Sigma-Aldrich, Copenhagen, Denmark. Treatment was carried out once a week. The mice were observed regularly; blood and urine samples were taken each week.

### 3.4. Induction of Disease in Mice

LS was induced in C57BL/6J mice with a single intraperitoneal injection of 25 mg/kg body weight bleomycin (Sigma-Aldrich). SLE was induced in NZB/NZW F1 mice with three intraperitoneal injections of tamoxifen (50 mg/kg body weight), with 1 day between the injections. After treatment, the animals were allowed to rest for 1 week and then treated as described in Section 3.3. Induced C57BL/6J and NZB/NZW F1 mice received intraperitoneal injections of dsD4 or dsD5 at a high dose (14 µg/dose, or 0.8 mg/kg body weight at an average body weight of 18 g) or a low dose (1.4 µg/dose, or 0.08 mg/kg body weight at an average body weight of 18 g).

In total, five mice were used per group; control animals received PBS instead of bleomycin and DNA antigens. The observation period lasted 5 weeks. During this time, the animals were weighed, and blood and urine were collected once a week. At the study termination point, the organs were split into two parts; one part was flash-frozen, and the other part was incubated in formalin for 48 h and then embedded in paraffin.

### 3.5. In-House ELISA of Mouse Samples

Whole blood from mice was centrifuged at 2000× *g* for 20 min at 4 °C. Serum was separated and stored at −80 °C.

The ELISA for nucleic acid antigens was carried out as described in Section 3.2.

PLox antigen was prepared by treating heart cardiolipin (Avanti Polar lipids, cat. no. 840012P, Alabaster, AL, USA) with 30% H_2_O_2_ in 60% MeOH, 5% CHCl_3_, and 35% MilliQ water. The product was purified using a 3K Amicon Ultra device (Millipore, Bedford, MA, USA) and confirmed with LC-MS.

The general protocol for the in-house PLox ELISA is given below. A Maxisorb 96-well plate was coated with PLox (8 µg/mL in ethanol, 100 µL/well) overnight at +4 °C. The plates were blocked with 1× PTB buffer (20 g BSA, 50 µL Tween-20, 1 L 1× PBS; 150 µL/well) at room temperature. Serum was diluted to a ratio of 1:100 in diluent (2 g BSA, 50 µL Tween-20, and 1 L 1× PBS; 100 µL/well), and the plate was incubated for 1.5 h at 37 °C. After four washes with 1× PBS containing 2% Tween 20, 100 µL of HRP-conjugated secondary antibody was added to each well; the plate was incubated at 37 °C for 1.5 h. After four washes, freshly prepared TMB solution was added to each well (3 mg of TMB, 5 mL of DMSO diluted to 50 mL with 0.1 M acetate buffer, pH 5.4–5.7, and 3 µL H_2_O_2_; 100 µL/well), and the plate was incubated for 20–30 min at room temperature. The reaction was stopped by adding 1 M H_2_SO_4_ (50 µL/well). The absorbance at 450 nm was measured in each well.

## 4. Conclusions

We investigated sequence-defined synthetic nucleic acid antigens in patients with LS and a bleomycin-induced LS mouse model. Based on the ELISA results, the serum samples from the patients with LS showed the highest binding of autoantibodies to dsD4, which contains the sequence of the human oncogene *BRAF*. In the serum from the patients with LS, the anti-dsD4 level affected the disease activity significantly. In the bleomycin-induced LS mice, dsD4 induced a strong polyclonal antibody response without worsening the course of LS. The lack of skin fibrosis in the mice may be due to the relatively low bleomycin dose, which could have blunted the full skin response. The serological response to dsD4 in LS was different from the response to dsD5, which was most active in the SLE mouse model used as a control. The IgG response to dsD4 increased stably over the 4-week observation after administering the last dose of antigen. Taken together, our data are the first to confirm the impact of nucleic acid sequences on the autoantibody response in LS in vivo. Our findings highlight the sequence specificity of the antibody response in LS and open the path to improved early-stage diagnostic tests.

## Figures and Tables

**Figure 1 ijms-24-17507-f001:**
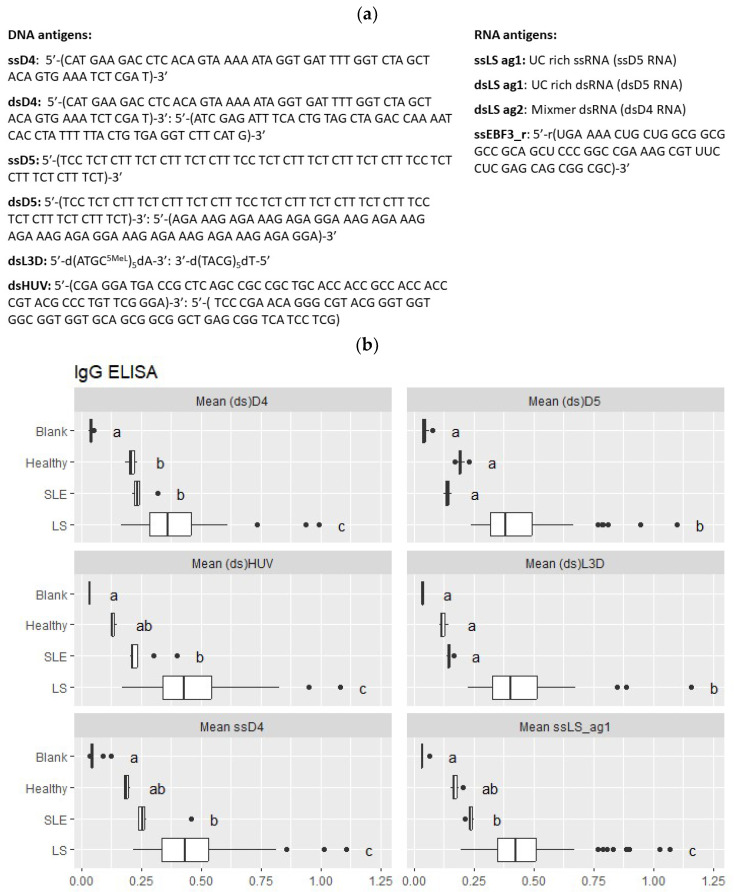
Evaluation of autoantibody reactivity to synthetic nucleic acid antigens. (**a**) Sequences of the evaluated DNA and RNA antigens. The enzyme-linked immunosorbent assay (ELISA) results are presented as boxplots for IgG (**b**), IgM (**c**), and IgE (**d**) for serum from patients with localized scleroderma (LS, n = 70), patients with systemic lupus erythematosus (SLE, n = 11), and healthy controls (n = 11). A different letter indicates a significant difference at the 0.05 level based on Tukey’s honest significant difference test (Appendix A).

**Figure 2 ijms-24-17507-f002:**
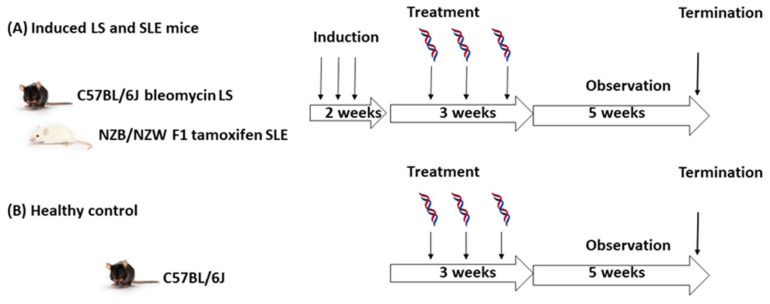
Overview of the in vivo study: (**A**) localized scleroderma (LS) induced in C57BL/6J mice (n = 20) and systemic lupus erythematosus (SLE) induced in NZD/NZW mice (n = 30); (**B**) C57BL/6J control mice (n = 30). LS was induced with a single intraperitoneal dose of 25 mg/kg body weight bleomycin. SLE was induced by three intraperitoneal injections of 50 mg/kg body weight of tamoxifen over the course of 1 week. The treatment was dsD4 or dsD5 delivered intraperitoneally at 1.4 or 14 µg/dose or phosphate-buffered saline (PBS, negative control). Each group contained five mice.

**Figure 3 ijms-24-17507-f003:**
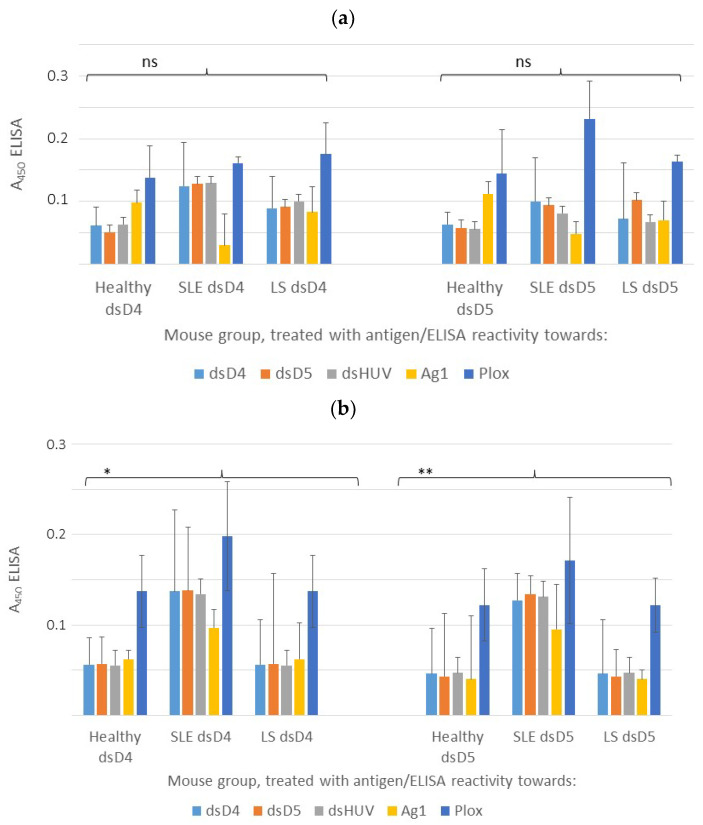
The enzyme-linked immunosorbent assay (ELISA) results for serum samples obtained from the in vivo study: IgG (**a**) and IgM (**b**). The data are presented for the end of the observation for the high-dose (H) treatment (14 µg/dose). The levels in the control samples (unimmunized/not treated healthy mice, average value for five mice) were as follows: for IgG, dsD4 = 0.03; dsD5 = 0.04; dsHUV = 0.05; Ag1 = 0.04; PLox = 0.1. For IgM, dsD4 = 0.03; dsD5 = 0.04; dsHUV = 0.04; Ag1 = 0.06; PLox = 0.1. The asterisks indicate the significance level of the *F*-test: ns = not significant; * = 5% significance; ** = 1% significance.

## Data Availability

Data is contained within the article and Appendix A.

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
