# Peer review of "Synthetic Nucleic Acid Antigens in Localized Scleroderma"

_ijms, 2023, doi:10.3390/ijms242417507_

Round 1

Reviewer 1 Report

Comments and Suggestions for Authors

I would like to thank the authors for there very nice and novel work. The introduction was very comprehensive and self explanatory. Results are a good to more research and action in this area.  

I was a bit puzzled regarding the aim, was it to compare between the autoantigen in LS and healthy control for better identification or compare it to SLE to differentiate

In line 208, it better to make un-immunized rather not immunized.   Comments on the Quality of English Language

English language was very good.

Author Response

We appreciate the Referee’s comments. The aim now has been clarified, page 2: “Supported by recent findings obtained using sequence-controlled antigens and human sera samples[14,15], herein we hypothesized that the nucleotide sequence of nucleic acid antigens has an impact on the autoimmune reactions in localized scleroderma. We tested our hypothesis in vitro and in vivo using human samples and mouse model of LS, and compared it to SLE- and healthy controls.”

Reviewer 2 Report

Comments and Suggestions for Authors

The manuscript is generally well written. There are some concerns

1) In figure 1 it would be helpful to include statistics in the figure.

In the text there is a statement "Not immunized mice did not show signs of inflammation after administration of antigens." This is unclear. What signs of inflammation? 

It is unclear why the tamoxifen induced NZB/NZW mice were used.

In figure 3 the data would be clearer if presented Healty/SLE and LS side by side for each antigen, making direct comparison easier. Inclusion of statistics would be helpful.

The uniqueness of the study gets lost. We have known about ssRNA, ssDNA, dsRNA, and dsDNA as antigens in SLE for a long time. Is the novelty the scleroderma? That should be more the focus.

Comments on the Quality of English Language

Some minor word usage issues, awkward wording.

Author Response

We appreciate the Referee's comments! Our response to each of them is given below.

1) In figure 1 it would be helpful to include statistics in the figure.

This has been done.

In the text there is a statement "Not immunized mice did not show signs of inflammation after administration of antigens." This is unclear. What signs of inflammation?

We appreciate the Referee’s comment. This has been corrected as follows:

“After administration of antigens, un-immunized mice did not show signs of scleroderma development in skin. However, they developed autoantibodies.”

It is unclear why the tamoxifen induced NZB/NZW mice were used.

Similarly to study of LS and SLE human samples that we describe in the paper, we used SLE mice as a control group. This has been clarified as follows:

“Similarly to using SLE control in human sample study, we also used tamoxifen-sensitized NZB/NZW F1 model of SLE as a control disease group for animal model investigation.”

Moreover, tamoxifen speeds up development of SLE and increases mice survival (Ann Rheum Dis. 2003 Apr;62(4):341-6).

In figure 3 the data would be clearer if presented Healty/SLE and LS side by side for each antigen, making direct comparison easier. Inclusion of statistics would be helpful.

This has been done.

The uniqueness of the study gets lost. We have known about ssRNA, ssDNA, dsRNA, and dsDNA as antigens in SLE for a long time. Is the novelty the scleroderma? That should be more the focus.

We appreciate the Referee’s comment. We included the following clarifications into Introduction:

“Supported by recent findings obtained using sequence-controlled antigens and human sera samples[14,15], herein we hypothesized that the nucleotide sequence of nucleic acid antigens has an impact on the autoimmune reactions in localized scleroderma. We tested our hypothesis in vitro and in vivo using human samples and mouse model of LS, and compared it to SLE- and healthy controls. To the best of our knowledge, this is the first study that investigates sequence specificity of nucleic acid antigens in LS in both human samples and animal model.”